# Explainable Self-Supervised Dynamic Neuroimaging Using Time Reversal

**DOI:** 10.3390/brainsci15010060

**Published:** 2025-01-11

**Authors:** Zafar Iqbal, Md. Mahfuzur Rahman, Usman Mahmood, Qasim Zia, Zening Fu, Vince D. Calhoun, Sergey Plis

**Affiliations:** 1Department of Computer Science, Georgia State University, Atlanta, GA 30302, USA; ziqbal5@student.gsu.edu (Z.I.); mahfuz.gsu@gmail.com (M.M.R.); qzia1@student.gsu.edu (Q.Z.); zfu@gsu.com (Z.F.); vcalhoun@gsu.edu (V.D.C.); 2Center for Translational Research in Neuroimaging and Data Science (TReNDS), Atlanta, GA 30303, USA; usman.mahmood134@gmail.com

**Keywords:** interpretability, explainability, schizophrenia, fMRI, pretraining, self-supervised, time reversal

## Abstract

Objective: Functional magnetic resonance imaging data pose significant challenges due to their inherently noisy and complex nature, making traditional statistical models less effective in capturing predictive features. While deep learning models offer superior performance through their non-linear capabilities, they often lack transparency, reducing trust in their predictions. This study introduces the Time Reversal (TR) pretraining method to address these challenges. TR aims to learn temporal dependencies in data, leveraging large datasets for pretraining and applying this knowledge to improve schizophrenia classification on smaller datasets. Methods: We pretrained an LSTM-based model with attention using the TR approach, focusing on learning the direction of time in fMRI data, achieving over 98 % accuracy on HCP and UK Biobank datasets. For downstream schizophrenia classification, TR-pretrained weights were transferred to models evaluated on FBIRN, COBRE, and B-SNIP datasets. Saliency maps were generated using Integrated Gradients (IG) to provide post hoc explanations for pretraining, while Earth Mover’s Distance (EMD) quantified the temporal dynamics of salient features in the downstream tasks. Results: TR pretraining significantly improved schizophrenia classification performance across all datasets: median AUC scores increased from 0.7958 to 0.8359 (FBIRN), 0.6825 to 0.7778 (COBRE), and 0.6341 to 0.7224 (B-SNIP). The saliency maps revealed more concentrated and biologically meaningful salient features along the time axis, aligning with the episodic nature of schizophrenia. TR consistently outperformed baseline pretraining methods, including OCP and PCL, in terms of AUC, balanced accuracy, and robustness. Conclusions: This study demonstrates the dual benefits of the TR method: enhanced predictive performance and improved interpretability. By aligning model predictions with meaningful temporal patterns in brain activity, TR bridges the gap between deep learning and clinical relevance. These findings emphasize the potential of explainable AI tools for aiding clinicians in diagnostics and treatment planning, especially in conditions characterized by disrupted temporal dynamics.

## 1. Introduction

Brain activity is spatially and temporally well-organized, even in resting-state conditions [1]. This inherent organization, assessed through resting-state functional connectivity, provides valuable insights into the brain’s intrinsic cognitive functions. Functional connectivity, identified using imaging techniques such as functional magnetic resonance imaging (fMRI), serves as a biomarker for various pathological conditions, including Schizophrenia, Alzheimer’s disease, and autism [2]. The fluctuations in the blood oxygenation level-dependent (BOLD) signal correspond to functionally relevant resting-state networks, which are captured by resting-state fMRI in the absence of goal-directed neuronal action [1]. However, interpreting the high-dimensional and noisy brain dynamics recorded by fMRI remains challenging.

Traditional Machine Learning models, while inherently interpretable, often struggle to capture the complex discriminative features and patterns associated with specific disorders. In contrast, deep learning frameworks can learn hierarchical representations directly from raw data through deep, layered architectures that facilitate the flow of information from input to output. However, this increased modeling capacity typically comes at the expense of interpretability [3]. This trade-off has motivated the development of explainable and interpretable Machine Learning methods, which aim to make predictions and decisions more intelligible to human users. Recent advancements in XAI techniques have been applied to rs-fMRI data, enabling the interpretation of brain connectivity patterns and their relationships to disorders [4,5]. These developments underscore the increasing need for transparent and interpretable deep learning frameworks in neuroimaging.

Explainability and interpretability, though often used interchangeably, have distinct meanings [6]. Interpretability refers to designing Machine Learning models that are inherently understandable, while explainability involves generating post hoc explanations for predictions from complex, black-box models [7]. As Artificial Intelligence-based systems find increasing applications in high-stake domains like healthcare, enhancing trust and transparency in decision making has become critical. This can be achieved either by designing inherently interpretable models or by employing model introspection methods to provide faithful explanations for complex models.

Interpretable models, such as logistic regression, linear models, and K-nearest neighbors, exhibit a clear relationship between inputs and outputs, making their predictions more understandable [8]. Standard machine learning (SML) techniques often fall under this category, relying on rules of inference to predict outcomes by leveraging linear and non-linear relationships between variables. While Standard Machine Learning methods offer transparency and are suitable for high-stake decision making, their predictive power is limited when data exhibits intricate relationships [9]. In such cases, more complex methods, including deep learning, are required.

For complex models, model-agnostic and model-specific post hoc techniques provide explanations for predictions. Model-agnostic methods, such as SHAP [10], LIME [11], and Perturbation, are applicable to any Machine Learning model. Model-specific methods, such as integrated gradients (IG) [12], gradient SHAP (GS) [10], and saliency maps [13], are tailored to specific model architectures. Example-based methods, including adversarial examples, influence functions, and counterfactual analysis, provide insights using specific examples from the dataset. Additionally, neural representation techniques, such as SVCCA [14], activation maximization [15], and TCAV [16], explore model internals to make latent representations interpretable.

Several studies have utilized these introspection techniques for time-series data. For example, Hugues Turbé et al. evaluated interpretability methods on ECG data, introducing new metrics—$AUC{\tilde{S}}_{top}$ and $F1\tilde{S}$—to quantify their relevance identification performance. Their findings emphasize that the effectiveness of these methods depends on both the dataset and the architecture [17]. Similarly, Rahman et al. applied IG and SmoothGrad integrated gradients to fMRI data using an LSTM-based architecture, demonstrating their efficacy in capturing salient features that improved classification performance across three neuromimaging datasets [3].

The methods discussed each have their own advantages and limitations, making them suitable for different applications. Adebayo et al. (2020) found that under standard conditions, gradient-based methods such as integrated gradients performed well in meeting end-user recommendations [18]. These methods effectively address common issues found in traditional saliency maps, including susceptibility to noise and sensitivity to minor input changes [19]. In our research, we observed that IG and GS demonstrated robustness, stability, and reduced sensitivity to noise across the datasets we analyzed [20]. Based on these findings, we selected IG for model analysis in our study.

This work aims to provide post hoc explanations for our previously proposed pretraining method, time reversal (TR) [21]. Designed for time-series data, the TR method reverses the order of time points in the data and pretrains a model on both forward and reversed time courses. This approach enables the model to effectively capture temporal information that can be leveraged in downstream tasks. In our previous work, we demonstrated the generalizability of the TR method across multiple datasets and diseases. Specifically, we applied it to two datasets for schizophrenia classification (FBIRN and COBRE), one dataset for autism classification (ABIDE [22]), and one dataset for Alzheimer’s disease classification (OASIS [23]). Models pretrained with time reversal significantly outperformed those trained from scratch, even when using fewer subjects, underscoring the cross-domain benefits of TR-based pretraining.

In this study, we focus on explaining model predictions during both pretraining and downstream schizophrenia classification. Schizophrenia is a psychiatric disorder often characterized as a progressive illness based on the degeneration of brain functions. Its symptoms include delusions, thought disorder, hallucinations, motor and cognitive impairment, and reduced expression of emotions, to name a few. Schizophrenia is episodic in nature and known as acute Schizophrenia. Patients have episodes of Schizophrenia during which their condition worsens followed by periods where there are few or no symptoms. Although the etiology of the illness remains largely unknown, a progressive decrease in the gray matter of the brain has been witnessed in adolescents in the early and chronic stages of the disease [24].

Leveraging an LSTM framework with an attention mechanism, we analyze how models learn temporal dependencies during pretraining and apply this knowledge to improve schizophrenia classification performance. Techniques like saliency maps highlight critical input features, while Earth Mover’s Distance (EMD) measures the focus of these maps, indicating the model’s attention to significant temporal features. Additionally, we use the area under the curve (AUC) score [25] to evaluate performance. These tools collectively bridge the complexity of deep learning with the need for transparency in high-stake applications like healthcare diagnostics.

In summary, this work aims to bridge the gap between high-performance deep learning models and the need for transparency in healthcare applications. By leveraging TR-based pretraining and advanced interpretability techniques, we provide a framework that is both robust and trustworthy for neuroimaging-based diagnostics.

## 2. Materials and Methods

The proposed methodology consists of four key steps: (1) pretraining with Time Reversal, (2) providing post hoc explanations for the pretraining phase using model introspection methods, (3) evaluating downstream classification performance of models with and without the Time Reversal-based pretraining, and (4) assessing the impact of pretraining on downstream tasks by analyzing the spikiness and uniformity of salient features along the time axis. We have discussed each of the four steps in the following subsections. A schematic description of the proposed methodology is shown in Figure 1.

### 2.1. Pretraining Method

#### 2.1.1. Problem Formulation and Objective

The Time Reversal-based pretraining strategy is designed to capture the temporal dynamics inherent in the data. Here is how the method works:

Let the fMRI dataset after independent component analysis (ICA) preprocessing to be represented as $X\in R^{T\times N}$, where *T* denotes the number of time points, and *N* represents the number of independent components or features. The time reversal operation is defined as a transformation R(·), which reverses the order of time points for each feature. Mathematically, the reversed dataset, $X_{\mathrm{reverse}}$, is given by the following:$$R\left(X\right)=X_{\mathrm{reverse}},\;\mathrm{where}\;X_{\mathrm{reverse}}\left[t,n\right]=X\left[T-t-1,n\right],\;\forall t\in \left[0,T-1\right],n\in \left[1,N\right].$$

This results in a new dataset $X_{\mathrm{reverse}}\in R^{T\times N}$, which retains the same dimensions as the original data *X*, but with time points reversed.

During the pretraining phase, the model is trained to distinguish between the original and time-reversed data. This combined dataset can be formulated as$$X_{\mathrm{combined}}=\left\{\left(X,y=0\right),\left(R\left(X\right),y=1\right)\right\},$$
where y=0 for the original time series and y=1 for the time-reversed series.

The objective of the model is to predict whether a given time series is the original or the reversed version. We define the model as $f_{\theta }\left(X\right)$, parameterized by θ, where the output $\hat{y}=f_{\theta }\left(X\right)$ represents the predicted probability that the input is time-reversed. The loss function for training the model can be expressed as$$ℒ\left(\theta \right)=-\frac{1}{2T}\sum_{\left(X_{i},y_{i}\right)\in X_{\mathrm{combined}}}\left[y_{i}\log\left(f_{\theta }\left(X_{i}\right)\right)+\left(1-y_{i}\right)\log\left(1-f_{\theta }\left(X_{i}\right)\right)\right].$$

The model learns latent representations $Z=g_{\theta }\left(X\right)$ through this pretraining process, where $g_{\theta }\left(\cdot \right)$ is the encoder network. These representations *Z* are then used for downstream tasks, such as classification or further interpretation.

#### 2.1.2. Temporal Dynamics in the Dataset

The dataset *X* contains temporal dependencies between time points, where each point $x_{t}$ is conditionally dependent on previous points $x_{t-1},x_{t-2},\ldots$. This relationship can be expressed as$$P\left(X\right)=P\left(x_{1}\right)\prod_{t=2}^{T}P\left(x_{t}|x_{t-1},\ldots ,x_{1}\right).$$

Reversing the time order disrupts these dependencies. For the time-reversed sequence $X_{\mathrm{reverse}}$, the conditional probability becomes$$P\left(X_{\mathrm{reverse}}\right)=P\left(x_{T}\right)\prod_{t=1}^{T-1}P\left(x_{T-t}|x_{T-t+1},\ldots ,x_{T}\right).$$

As a result, the temporal transitions and dependencies in $X_{\mathrm{reverse}}$ differ significantly from those in *X*. To distinguish between the two, the model must learn to encode the temporal structure of the data effectively.

#### 2.1.3. Expanded Theoretical Analysis

The core idea behind the Time Reversal pretraining strategy is that reversing the temporal sequence disrupts the natural order of events, requiring the model to extract and leverage temporal patterns for successful classification. This process encourages the encoder $g_{\theta }\left(\cdot \right)$ to learn robust representations *Z* that encode time-sensitive features, such as transitions and causal dependencies.

By optimizing the binary cross-entropy loss, the model minimizes the uncertainty H(y∣X) in predicting whether a sequence is reversed. This inherently maximizes the mutual information I(X;y) between the input *X* and the output label *y*, aligning the learned representations with the underlying temporal dynamics of the data.

The effectiveness of this approach aligns with principles of self-supervised learning, where the auxiliary task of identifying reversed sequences serves as a proxy for uncovering latent temporal structures. In Section 3, we provide experimental evidence to substantiate these claims.

### 2.2. Datasets

We used one synthetic dataset and two neuroimaging datasets for pretraining and post hoc explainability, and employed three additional datasets for the downstream classification tasks. All neuroimaging datasets are preprocessed using a technique called independent component analysis [26]. The details of preprocessing are presented in Section 2.3.

#### 2.2.1. Synthetic Dataset for Pretraining

The synthetic data consists of concatenated chirp signals designed to match the dimensions of the real datasets (i.e., 53 components with each component comprised 1200 time points). A chirp is a signal in which frequency decreases (down-chirp) or increases (up-chirp) with time. For generating the synthetic data, we used up-chirp signals with varying start and end frequency values. The chirp signal was chosen to generate time series data by incorporating a time component and capturing the changes associated with it. The employment of synthetic chirp signals to model time-series data in neuroimaging research, while not directly mimicking biological signals, offers significant advantages in understanding brain dynamics. Synthetic chirps can simulate frequency modulations akin to those observed in neural oscillations during cognitive tasks, providing a controlled environment to study how neural networks process these changes. They enable the isolation of specific signal characteristics for algorithm development and testing, serving as benchmarks to assess the performance of a model. By augmenting real data or providing a training ground for educational purposes, chirp signals contribute to the development of more robust models that can generalize to the complexities of biological data, thereby enhancing our ability to interpret real-world neuroimaging signals.

#### 2.2.2. Neuroimaging Datasets for Pretraining

In the next phase, we used healthy controls from two publicly available datasets from the Human Connectome Project (HCP) [27] and the UK Biobank [28]. The HCP and UK Biobank datasets each contain 823 healthy control subjects. Each subject is represented by 53 non-noise components. In the HCP dataset, there are 1200 time points per subject (53 components × 1200 time points), while in the UK Biobank dataset, there are 490 time points per subject (53 components × 490 time points). The purpose of pretraining on healthy controls is to learn prior knowledge about the general signal dynamics from large datasets.

#### 2.2.3. Neuroimaging Datasets for Downstream Classification

The three datasets utilized for downstream classification are related to schizophrenia. Specifically, we employed data from the following projects: the Function Biomedical Informatics Research Network (FBIRN) [29], which includes 311 subjects (160 patients and 151 healthy controls); the Centre of Biomedical Research Excellence (COBRE) [30], comprising 157 subjects (89 patients and 68 healthy controls); and the Bipolar-Schizophrenia Network for Intermediate Phenotypes (B-SNIP) [31], consisting of 589 subjects (251 patients and 338 healthy controls).

### 2.3. Preprocessing

The raw resting-state fMRI data are processed using statistical parametric mapping (SPM) in MATLAB 2016. The first five scans were discarded to allow for signal equilibrium and participants’ adaptation to the scanner’s noise. Motion correction was performed using the rigid body motion correction toolbox in SPM to address subject head movements, followed by slice-timing correction to account for temporal differences in slice acquisition. Subsequently, the fMRI data were normalized to the Montreal Neurological Institute (MNI) standard space using an echo-planar imaging (EPI) template. The fMRI data are resampled to 3×3×3 mm^3^ and smoothed with a Gaussian kernel having a full width at half maximum (FWHM) of 6mm. Subjects with head motion exceeding 3 degrees of rotation or 3 mm of translation were excluded from further analysis. To ensure high data quality, quality control (QC) was performed on the spatial normalization output, and subjects with limited brain coverage were excluded [32]. ICA time courses were utilized, as they provide a more robust representation of fMRI data compared to anatomical or atlas-based approaches [33]. For each dataset, we employed a fully automated framework for extracting components. To generate network templates, spatial group ICA was conducted on two independent datasets, namely the Human Connectome Project (HCP) and the Genomics Superstruct Project (GSP). For each dataset, group-level ICs were estimated and matched by comparing their spatial maps. IC pairs with a spatial correlation of ≥0.4 were considered consistent and reproducible across datasets. A subset of these matched ICs was classified as intrinsic connectivity networks (ICNs), excluding components related to physiological noise, motion artifacts, or imaging artifacts. This classification was performed by five fMRI experts, who evaluated ICs based on specific criteria: activation peaks in gray matter, low spatial overlap with vascular, ventricular, or motion-related artifacts, and dominant low-frequency fluctuations in their corresponding time courses. ICs receiving at least three votes from the experts were deemed meaningful ICNs. These ICNs were subsequently used as templates for individual-level ICA analysis. Using the same procedure, 100 ICA components were estimated for each dataset. For the current study, we focused on 53 intrinsic networks that perfectly matched the standard network templates [3].

### 2.4. Model Architecture and Training Methodology

The proposed architecture is designed for processing sequential data with a focus on temporal dynamics and feature importance. It begins with an LSTM encoder to capture dependencies across time, transforming the input sequence into a series of hidden states. A custom attention mechanism follows, where the last hidden state is expanded and concatenated with all LSTM outputs to compute attention weights through two linear layers. These weights, after softmax normalization, are used to emphasize critical temporal features, creating an attention-weighted context vector. This vector is then processed by a decoder, consisting of linear layers with dropout for regularization, culminating in a sigmoid activation for binary classification tasks. The model’s weights are initialized using Xavier normal initialization, aiming to enhance training stability, particularly for the encoder, attention, and decoder components. A visual representation of the network architecture is shown in Figure 2. We used the same architecture and same hyperparameters for pretraining and the downstream prediction tasks.

For our experiments, a learning rate of $7\times 10^{-4}$ was selected after testing various values in the range $1\times 10^{-3}$ to $1\times 10^{-5}$. This value provided stable convergence without oscillations or premature stagnation. The Adam optimizer, combined with the ReduceLROnPlateau scheduler, dynamically adjusted the learning rate based on validation loss, ensuring effective optimization. A batch size of 32 was chosen based on empirical testing, balancing computational efficiency and gradient stability. Smaller batch sizes resulted in noisier gradient updates, while larger sizes slowed convergence due to fewer updates per epoch. To ensure robust evaluation, we employed 10-fold cross-validation, which reduced the risk of overfitting to specific data splits and provided a comprehensive assessment of the model’s generalization capabilities.

Training was conducted with a maximum of 1000 epochs, and early stopping was applied to prevent overfitting. Early stopping monitored the validation loss, halting training when no improvement was observed for a specified number of epochs, thus ensuring that the model did not overtrain. Convergence during training was verified by monitoring the loss curves, which exhibited smooth declines for both training and validation, with no signs of divergence or overfitting. Metrics such as accuracy and AUC consistently improved during training, further confirming stable convergence. To evaluate the contributions of individual model components, ablation studies were performed. Removing the attention mechanism led to a significant drop in performance, demonstrating its role in emphasizing critical temporal features. Disabling dropout regularization caused overfitting, evident from a larger gap between training and validation performance. Replacing the LSTM encoder with a GRU resulted in slightly reduced performance, underscoring the LSTM’s ability to capture temporal dependencies effectively. These results validate the robustness of the architecture and the importance of its components in achieving optimal performance.

### 2.5. Evaluation Metrics and Interpretability Framework

In this work, we aim to interpret and provide faithful post hoc explanations of the pretraining method (Time Reversal). An interpretation is a translation between two domains such that the concepts of the first domain can be understood in terms of concepts of the second domain. Here, we are interested in interpretations of a neural network in terms of human-understandable representations. To achieve the objective outlined above, we investigated various attribution methods documented in the literature. We discovered that gradient-based techniques, specifically integrated gradients, performed most effectively in our experiments.

To evaluate the impact of pretraining on downstream tasks, we utilized three datasets: FBIRN, COBRE, and B-SNIP. For each of these datasets, we applied pretrained weights from the HCP dataset for Schizophrenia classification. Additionally, we trained models from scratch to enable a direct comparison of models’ performance with and without pretraining. To assess the performance of the models, we used AUC as our evaluation metric because it provides a robust measure of model performance by considering the true positive rate and false negative rate across various cutoff thresholds. AUC is especially preferred over accuracy in the context of imbalanced datasets, such as those used in our schizophrenia classification task, as it offers a reliable evaluation across varying thresholds and is widely regarded as a dependable performance indicator in such scenarios.

We further sought to provide explanations for the improved predictive performance in Schizophrenia classification tasks with pretraining by using EMD. EMD calculates the minimum “cost” required to transform one distribution into another. It quantifies the dissimilarity between two frequency distributions by measuring the minimal amount of work needed to match one distribution to another, based on a fixed reference template [34]. In simple terms, Earth Mover’s Distance is like figuring out how much effort it would take to rearrange one pile of sand to look exactly like another pile. If you imagine each pile representing how data are spread out, EMD measures the “work” or “cost” of moving grains from one pile to match the shape and size of the other pile. This cost depends on how far and how much sand needs to be moved.

Our objective in employing EMD was to evaluate the “spikiness” of the resultant distributions from saliency maps. EMD was chosen for its unique capability to measure the distance between two probability distributions over a given metric space, which, in our context, represents the temporal distribution of feature importance across time series data from resting-state fMRI. Unlike other metrics that might focus on local differences or aggregate statistics, EMD provides a holistic view of how the importance of features shifts over time, offering insights into the temporal dynamics critical for understanding neural processes. This metric’s ability to capture the overall ’movement’ or shift in importance across the entire time axis allows for a more nuanced interpretation of how our model prioritizes different temporal features, potentially correlating with known neurobiological mechanisms of the disease.

In this study, a “distribution” refers to the flattened saliency maps along the time axis, transforming, for instance, a 53 × 140 saliency map into a linear distribution of 140 time points. The spikiness measure we derive from this indicates the importance of each time point in the model’s prediction, where a distribution with pronounced peaks or “spikes” implies that fewer, more specific time points are crucial for the prediction, leading to a lower EMD value that signifies higher variability and focus. ely, Conversa smoother distribution with less pronounced peaks suggests a more diffused importance across time points, resulting in a higher EMD value, indicating less variability. This approach aids in understanding how the model prioritizes different time segments in its decision-making process, as visualized in Figure 3, where the contrast in spike distributions between two different conditions or models is highlighted. This visualization underscores the interpretative power of EMD in distinguishing how models focus on temporal features in rs-fMRI data for schizophrenia classification.

## 3. Results

We present the results in two main sections: first, we provide post hoc explanations for the pretraining method using saliency-based techniques; second, we evaluate the impact of pretraining on the downstream schizophrenia classification task, complemented by explanations for the downstream tasks using Earth Mover’s Distance scores.

### 3.1. Post Hoc Explanations for the Pretraining Method

Before presenting the results, it is important to first explain how the saliency maps for each subject are generated and displayed. As illustrated in Figure 4, we pass the time series data (in both forward order, $T_{1},T_{2},T_{3},\ldots ,T_{n}$, and reverse order, $T_{n},\ldots ,T_{3},T_{2},T_{1}$) and the pretrained model to the model introspection algorithm (integrated gradients) to produce the corresponding saliency maps. These saliency maps are represented as forward and reverse sequences: forward saliency maps, $S_{1},S_{2},S_{3},\ldots ,S_{n}$, and reverse saliency maps, $S_{n},\ldots ,S_{3},S_{2},S_{1}$. To facilitate a comparison, we flip only the reversed saliency maps and visualize the resulting matrices, given that we have 53 vectors, each corresponding to a time point. The purpose of this procedure is to align the salient features across both the forward and reverse saliency maps, ensuring that the location of key features on the time axis is consistent in both orientations. Figure 5 illustrates the saliency map for a single subject, where time courses are plotted along the x-axis and ICA-extracted components along the y-axis. The vertical bars, marked with red rectangles, denote the top 5% most salient features identified using the integrated gradients’ technique, highlighting their significant impact on the model’s predictions. The samples depicted in subsequent saliency maps Figures exhibit the same pattern as demonstrated in Figure 5.

The resulting saliency maps of the pretrained model applied to the synthetic data are shown in Figure 6. The red vertical bars highlight the most significant features that contributed to learning the direction of time. Notably, these bars are aligned across both the forward and reverse saliency maps. Our analysis suggests that the location of certain features helps the model discern the direction of time in the time series data. Since we used up-chirp signals, the frequency increases from left to right, which is reflected in the saliency patterns. It is also worth noting that the salient regions are predominantly concentrated on the low-frequency side of the chirp signal.

Next, we shifted our focus to the neuroimaging datasets to investigate whether the trend observed with the synthetic data held true in real-world datasets. Specifically, we examined healthy controls from two datasets: the HCP (Human Connectome Project) and the UK Biobank.

When we generated saliency maps for the test sets from both the HCP and UK Biobank datasets, we observed a similar phenomenon—alignment of the vertical bars across the forward and reverse saliency maps. However, for some subjects, this alignment was not apparent. To quantify the degree of alignment, we calculated the Pearson correlation coefficients between the features of the forward and reverse ordered data. The correlation coefficient is a statistical measure that represents the degree of linear association between two variables [35].

The correlation coefficient serves as an indicator of the alignment between the forward and reverse saliency maps. We found that in both datasets, over 75% of the subjects had correlation coefficients greater than 0.55 (as shown in Figure 7), suggesting that for the majority of subjects in both test sets, the salient features aligned in both forward and reverse maps along the time axis.

We present the five subjects with the highest correlation values for each dataset (HCP and UK Biobank) in Figure 8. For most of these subjects, the vertical bars are well aligned. While a few exceptions exist, these are likely outliers and can be ignored. Conversely, the subjects with the lowest correlation values are shown in Figure 9. For the subjects in the top half of this group, no alignment is observed, but as we move toward the lower half, where the correlation values are higher, some alignment is evident.

The alignment of forward and reverse saliency maps significantly enhances interpretability by providing a robust mechanism to validate the reliability of the extracted salient features across time. When saliency maps for forward and reverse time points are aligned, it demonstrates that the most influential features identified by the model are consistent and invariant to the temporal direction of the input. This consistency indicates that the features are intrinsic to the data and not an artifact of the model’s training process or temporal biases. By flipping the reverse saliency maps and observing their alignment with the forward saliency maps, we gain confidence in the model’s ability to capture meaningful patterns that are not contingent on the directionality of time. Moreover, this alignment helps to visually validate the temporal importance of features, enhancing the interpretability of how the model identifies key patterns over time. This process is particularly useful in domains like neuroscience, where identifying temporally relevant features is crucial for understanding underlying physiological processes.

As an additional validation, we employed a model-agnostic method called *submodular pick* to identify representative subjects from the test sets. This algorithm, proposed by Ribeiro et al. [11], provides a global understanding of the model by selecting subjects that best represent the overall behavior of the model. We fed flattened saliency maps into the submodular pick algorithm, which then selected a set of representative subjects. The attributions for these representative subjects are shown in Figure 10. The high correlation values and alignment of the vertical bars in the majority of these representative subjects further support our argument that the matching location of salient features along the time axis plays a crucial role in the model’s decision-making process.

This alignment is significant because it provides insight into how the model learns to determine the direction of time. By identifying a subset of significant features and learning their locations, the model is able to infer the direction of time and, in turn, capture the temporal dependencies within the data.

### 3.2. Downstream Tasks: Performance and Explanations

#### 3.2.1. Performance on the Schizophrenia Classification Tasks

We evaluated the performance of pretraining on the Schizophrenia classification task across three different datasets, using two distinct architectures: the proposed architecture (LSTM + attention) and the wholeMILC model [36]. The results, presented in terms of AUC scores, are shown in Figure 11, with performance measured using 10-fold cross-validation.

It is important to highlight that the proposed architecture consists of a unidirectional LSTM followed by an attention mechanism, whereas wholeMILC employs a more complex design. Specifically, wholeMILC uses a convolutional neural network encoder to process the time series data in a sequence of windows, applies attention to each encoded window, and then employs a bidirectional LSTM with an additional attention mechanism on top of it. The proposed architecture has fewer parameters compared to wholeMILC, making it less complex and more interpretable.

The results demonstrate that pretraining significantly improves model performance across all three datasets, outperforming models that were trained solely on the downstream datasets. Interestingly, wholeMILC performed better than the proposed architecture when no pretraining was applied, suggesting that more complex models may achieve higher performance but at the cost of reduced interpretability. However, when pretraining with time reversal was applied, the performance of the proposed architecture was comparable to state-of-the-art results, as reported by Pavel Popov et al. [37], despite using a simpler, more interpretable architecture. This finding highlights that with pretraining, even a less complex model can achieve competitive results, all while maintaining better explainability.

#### 3.2.2. Explanations for the Schizophrenia Classification Tasks

To assess the impact of pretraining on the patients in the downstream task, we calculated saliency maps both with and without pretraining weights. We then selected the top 5% of the most salient points, flattened them, and plotted the results. These plots, representing eight subjects from each of the three downstream datasets, are shown in Figure 12.

We observed that, in most cases, models using pretraining weights exhibited fewer spikes in their saliency maps. In contrast, without pretraining, the salient features were more scattered across the time axis. However, we did observe some subjects where the opposite trend occurred. To identify a dominant pattern, we used the Earth Mover’s Distance metric, which quantifies the “spikiness” or “scatteredness” of the salient points along the time axis.

The EMD box-and-whisker plots for Schizophrenia patients in the downstream tasks are shown in Figure 13. The higher EMD values for the models with pretraining indicate that the saliency points are concentrated into fewer spikes, suggesting that the model identified discriminative activity in shorter, more focused time intervals. This aligns with findings from a similar study on Schizophrenia classification (FBIRN) [3], which reported similar patterns of concentration in saliency after pretraining.

In summary, the EMD results highlight that pretraining encourages the model to focus on specific, discriminative time windows in patients, leading to more concentrated and interpretable saliency patterns, consistent with prior research in the field.

### 3.3. Comparison of Time Reversal with Other Pretraining Methods

We compared our Time Reversal pretraining approach with two other methods: order-contrastive pretraining (OCP) [38] and permutation contrastive learning (PCL) [39]. Similar to our Time Reversal technique, both OCP and PCL leverage the temporal sequence of time series data in a self-supervised manner to extract meaningful temporal information. OCP involves sampling pairs of time segments from each trajectory in the input data. For half of these pairs, the order is switched to create negative examples, while the other half remains in their correct temporal sequence, forming positive examples. The model is then trained to predict whether a given pair is in the correct order or not, effectively learning to discern temporal dependencies. On the other hand, PCL also uses positive pairs that are consecutive windows in the correct order but differs in its approach to negative sampling. Here, negatives are random window pairs from the same trajectory which might still be in the correct temporal order, contrasting with OCP where negatives are always misordered. This key difference in negative sampling might influence how each method learns from and represents temporal structures, with OCP potentially offering a more explicit signal about order due to its deliberate misordering of negatives.

The models are pretrained using Time Reversal, OCP, and PCL, and the effectiveness of the pretraining is evaluated on a downstream Schizophrenia classification task using the FBIRN dataset. The results are shown in Figure 14. In our comparative analysis, Time Reversal demonstrated superior performance over both OCP and PCL across two key metrics: AUC and balanced accuracy (BA). For AUC, Time Reversal exhibited a higher median performance (approximately 0.9172) compared to OCP (around 0.8057) and PCL (about 0.8708), suggesting a more robust capability in distinguishing between classes across various thresholds. This was mirrored in the consistency of Time Reversal’s results, with a tighter interquartile range indicating less variability. In terms of balanced accuracy, Time Reversal again led with a median of roughly 0.9238, showcasing its ability to handle class imbalances effectively by maintaining high accuracy for both true positives and true negatives. OCP, with a median of approximately 0.9128, showed slightly more variability, while PCL had a median around 0.8965 with a notably wider spread, including significant outliers. These results highlight Time Reversal’s robustness and consistency in leveraging temporal information for enhanced model performance in time series data analysis.

## 4. Discussion

Functional magnetic resonance imaging (fMRI) is preferred over other imaging modalities owing to its non-invasive nature. However, the high-dimensional nature (4D) and complicated relationship among features limit the use of standard Machine Learning algorithms for data analysis, despite them being inherently interpretable. Deep neural networks such as convolutional neural networks, recurrent neural networks, transformers, etc., on the other hand, have the capability to automatically extract features and exploit the information available from minimally preprocessed data to identify the subtle patterns and discriminative representations within the data [9]. This complexity, however, gives rise to black-box models that lack interpretability and hence make them less trustworthy, especially in high-stake domains such as neuroimaging. Luckily, we have model introspection methods available that can be used to provide post hoc explanations to the predictions of a deep learning model. The understanding of the internals of a model and the rationale behind a decision helps trust a model’s decision. This work demonstrates that it is possible to achieve high predictive performance out of a model and simultaneously provide rationale behind predictions of a deep framework to make it clinically relevant.

Deep neural networks require a lot of data for efficient training. Data scarcity in the medical domain is very prevalent due to patient data laws. One of the solutions to this problem is to use efficient pretraining methods that are capable of working with less data and still produce acceptable results in terms of performance. We use a pretraining method called Time Reversal to pretrain a model on different larger datasets. The purpose of this pretraining is to learn general structures and temporal information in the data. We pretrained two models to learn the order of time points in two different datasets (HCP, UK Biobank) with more than 98% accuracy.

The next step was to use IG to generate saliency maps on the test datasets. We experimented with different attribution algorithms and found out that IG’ attributions were more interpretable given the datasets and network architecture we used. We observed the alignment of vertical bars in forward and reverse time points of the saliency maps in the majority of the subjects. The vertical bars represent the most salient features that the model considered to make a prediction. We analyze that the model picked a subset of salient features in the data and based on the location of these features on the time axis, the model was able to learn the order of time courses. These post hoc explanations correspond to the working of a human brain. If a person wants to know the order of some objects, the brain would focus on the location of one or a subset of objects to reach a conclusion.

To evaluate the effectiveness of Time Reversal in downstream tasks, we applied it to three Schizophrenia-related datasets: FBIRN, B-SNIP, and COBRE. The results demonstrate that transferring pretrained weights to downstream classification tasks provides a significant performance boost, as evidenced by improved AUC scores. Pretraining proved beneficial across all datasets, despite their varying age ranges [21]. The advantages of transfer learning are especially pronounced in the COBRE dataset, which has a limited number of subjects (only 157), making it more challenging to train and test the model. As shown in our previous work [21], even using only one-third of the available data for training, pretraining still enhanced model performance compared to training the model solely on the downstream data.

To investigate the effectiveness of Time Reversal in downstream tasks beyond predictive performance, we evaluated the attributions of models trained with and without pretraining. We selected the top 5% of the most salient points, aggregated them along the components axis, and plotted the results. The plots revealed one or more spikes of varying height. To quantify the “spikiness” of these patterns, we employed EMD, a metric that estimates the spread of features along the time axis. The patterns observed in the EMD scores reflect the concentration and spread of salient features along the time axis, which is crucial in understanding the temporal structure of brain activity in the context of schizophrenia. When pretrained weights were used, we found that salient features were more concentrated in smaller regions along the time axis, indicating that the model identified focused intervals of activity. This aligns with the episodic nature of schizophrenia, where specific periods of neural activity are more pronounced, and the brain’s temporal dynamics become more defined.

Interestingly, this concentration of salient features in shorter, contributing to more focused intervals being able to correspond to the brain’s processing of certain stimuli or cognitive tasks, which can be more disrupted or appear in a more fragmented manner people with in schizophrenia. The ability of our method to detect these more localized intervals is reflective of a potential shift in temporal dynamics in patients with schizophrenia, where cognitive processes may be more disrupted or decoupled in terms of timing.

We also demonstrated that Time Reversal consistently outperformed the baseline pretraining methods (OCP and PCL) in both AUC and balanced accuracy, demonstrating higher median performance and less variability. This suggests that Time Reversal captures temporal dependencies more effectively, leading to more reliable and robust models for time series data.

Moreover, when compared against other pretraining techniques, like PCL and OCP, TR has shown superior performance, highlighting its ability to better capture temporal dependencies critical for schizophrenia classification. This advantage underscores TR’s effectiveness in dealing with the inherent noise and complexity of rs-fMRI data.

The very high predictive accuracy during pretraining, along with the highly aligned salient features, provides insights into the model’s reliance on temporal order for its decision making. These maps substantiate that alignment cannot be achieved without accurately pinpointing the temporal dynamics in both directions. Moreover, results on real datasets for downstream tasks reveal the additional advantages of pretraining. In particular, the models were able to identify smaller subsets of features compared to their without pretrained versions, as reflected in Figure 12 and Figure 13. This phenomenon potentially indicates the model’s deeper understanding of the temporal dynamics within the disease signal.

While the manuscript emphasizes the importance of interpretability and high predictive performance for schizophrenia classification, additional considerations for clinical integration of the proposed method could enhance its impact. Saliency maps, generated using IG, can play a crucial role in clinical decision making by offering a transparent understanding of the model’s reasoning process. For instance, the alignment of salient features with biologically meaningful temporal patterns could aid clinicians in identifying specific neural signatures associated with schizophrenia, supporting more precise diagnostics.

Moreover, the episodic nature of schizophrenia, as reflected in our findings on the temporal dynamics of the BOLD signal, highlights potential use cases where our method would be particularly impactful. For example, the ability to pinpoint shorter, more concentrated intervals of activity could inform the timing and targeting of interventions, such as cognitive behavioral therapy or pharmacological treatments, which may be more effective during these critical periods.

To integrate this method into clinical workflows, pretrained models could be deployed as part of a decision-support system in clinical imaging centers. These systems would analyze rs-fMRI scans and provide interpretable outputs, such as salient feature maps, alongside traditional diagnostic reports. Such maps could aid clinicians in verifying and contextualizing the model’s predictions by highlighting specific neural patterns associated with schizophrenia, enabling more informed and precise diagnostic decisions. Furthermore, the identification of salient temporal features could guide the timing of therapeutic interventions, such as cognitive behavioral therapy or pharmacological treatments, tailored to address the episodic nature of the disorder. Further research could explore whether these salient features align with known clinical biomarkers or predict treatment outcomes, thereby enhancing the practical utility and clinical relevance of this method.

This study significantly contributes to the growing field of explainable AI in neuroimaging by demonstrating that deep learning models can achieve high predictive performance while providing interpretable insights into their decision-making processes. The use of integrated gradients to generate saliency maps bridges the gap between model outputs and biological relevance, fostering trust in AI applications for clinical diagnostics. By aligning salient features with meaningful temporal dynamics, our approach highlights the potential for AI-driven tools to assist clinicians in identifying neural patterns associated with schizophrenia, supporting more precise and personalized treatment strategies. Moreover, this methodology can serve as a foundation for extending explainable AI techniques to other neurological conditions, enhancing their applicability and clinical impact.

While our proposed method demonstrates robust performance, it has certain limitations. First, regarding scalability, the computational demands of the Time Reversal pretraining task increase significantly with larger datasets, potentially limiting its applicability in real-time systems or environments with limited computational resources. Second, the datasets used for evaluation—FBIRN, COBRE, and B-SNIP—although well established in Schizophrenia research, may introduce biases due to their specific collection settings and population characteristics. These biases could impact the generalizability of the results across broader clinical or real-world settings.

To address scalability, future works could explore more efficient model designs, such as lightweight architectures or optimization techniques like knowledge distillation, to reduce computational overhead. Additionally, applying distributed training methods could facilitate scalability to larger datasets. Beyond Schizophrenia, evaluating the proposed method on datasets from other conditions, such as autism and Alzheimer’s disease, would broaden its applicability and validate its robustness in diverse clinical contexts. To mitigate potential biases, future research should include evaluations on more diverse and heterogeneous datasets, ideally collected from multiple institutions or geographical regions, to ensure the generalizability of the findings.

## 5. Conclusions

In conclusion, our study has demonstrated the superior efficacy of the time reversal pretraining method in enhancing both the performance and interpretability of deep learning models for schizophrenia classification using resting-state functional MRI data. By applying TR to datasets from the Human Connectome Project and UK Biobank for pretraining, followed by testing on the FBIRN, COBRE, and B-SNIP datasets, we observed significant improvements in median AUC scores, rising from 0.7958 to 0.8359 on FBIRN, 0.6825 to 0.7778 on COBRE, and 0.6341 to 0.7224 on B-SNIP compared to models without pretraining. Furthermore, when benchmarked against other pretraining strategies, like PCL and OCP, TR exhibited higher median AUC scores (0.8359 on FBIRN vs. 0.8057 for OCP and 0.8708 for PCL), underscoring its effectiveness in capturing critical temporal dependencies. This approach not only surpasses simpler models without pretraining but also competes well with complex models like wholeMILC, particularly when not pretrained, while offering enhanced interpretability through saliency maps and Earth Mover’s Distance. These findings lay a groundwork for developing more trustworthy and clinically applicable AI models in neuroimaging, potentially influencing diagnostic and therapeutic strategies for schizophrenia and extending to other neuropsychiatric conditions. 

## Figures and Tables

**Figure 1 brainsci-15-00060-f001:**
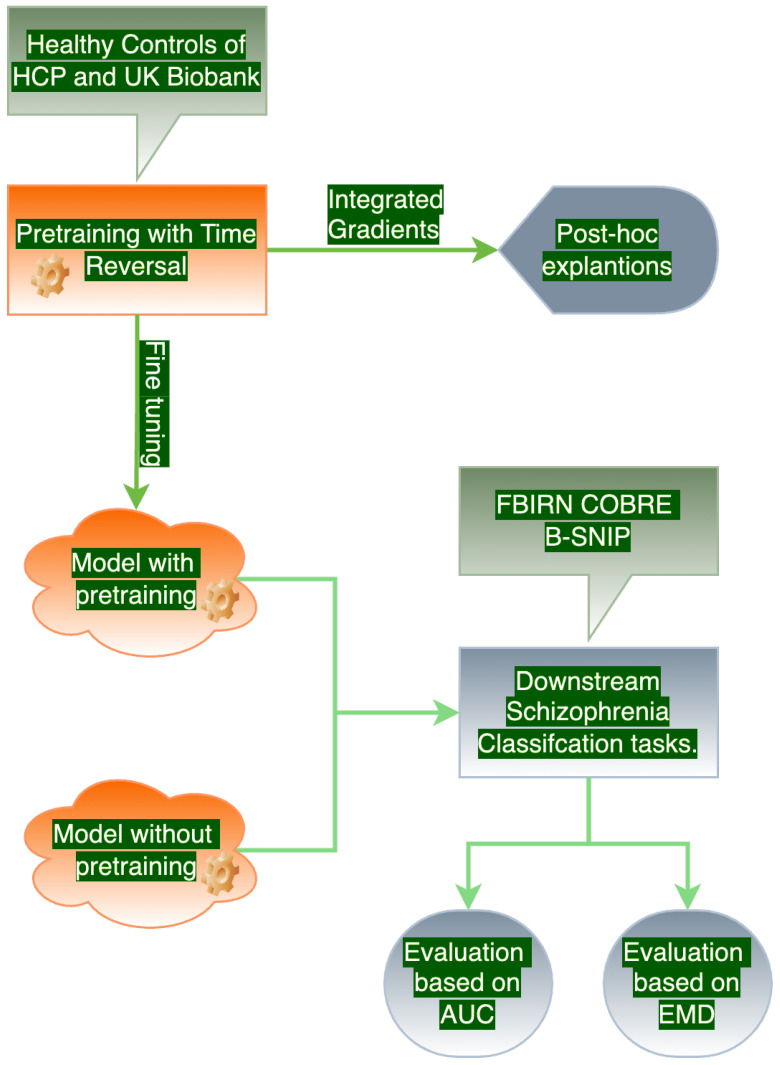
An overview of our proposed experimental setup.

**Figure 2 brainsci-15-00060-f002:**
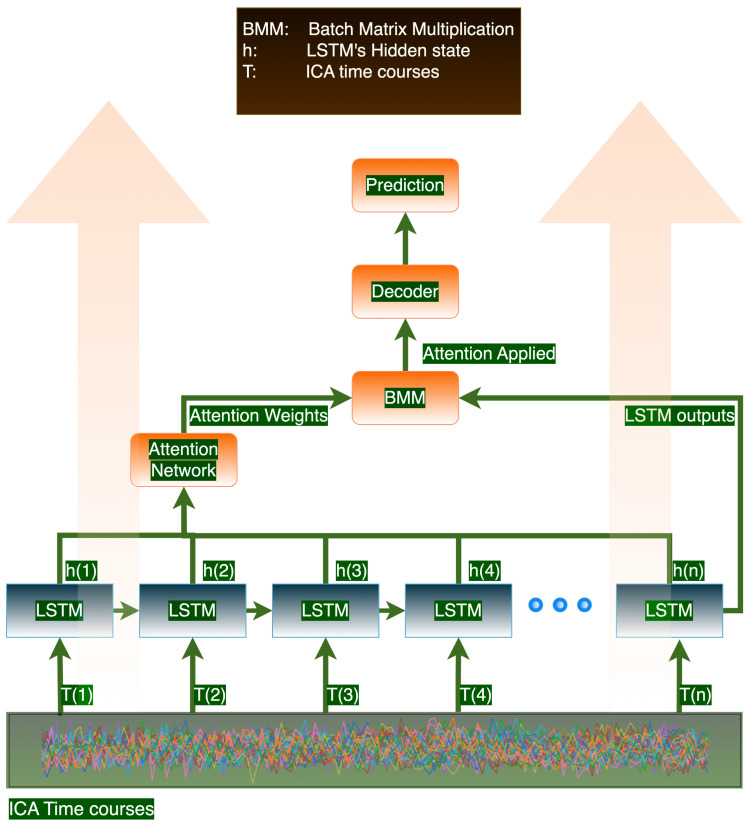
Proposed Architecture: We input time series data into an LSTM layer and apply an attention mechanism to selectively retain information from previous LSTM cells. This approach is especially beneficial when the sequence length is long.

**Figure 3 brainsci-15-00060-f003:**
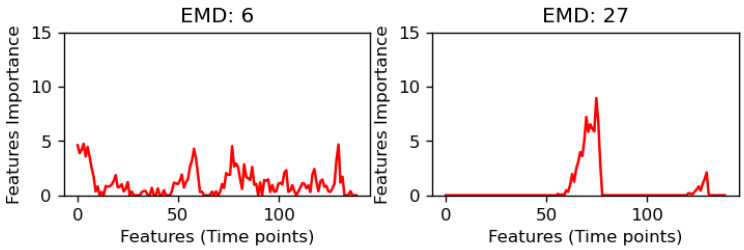
This figure illustrates the relationship between EMD and the distribution of spikes. As shown by the two distributions, fewer spikes correspond to higher EMD values, while more spikes and a more scattered distribution result in smaller EMD values.

**Figure 4 brainsci-15-00060-f004:**
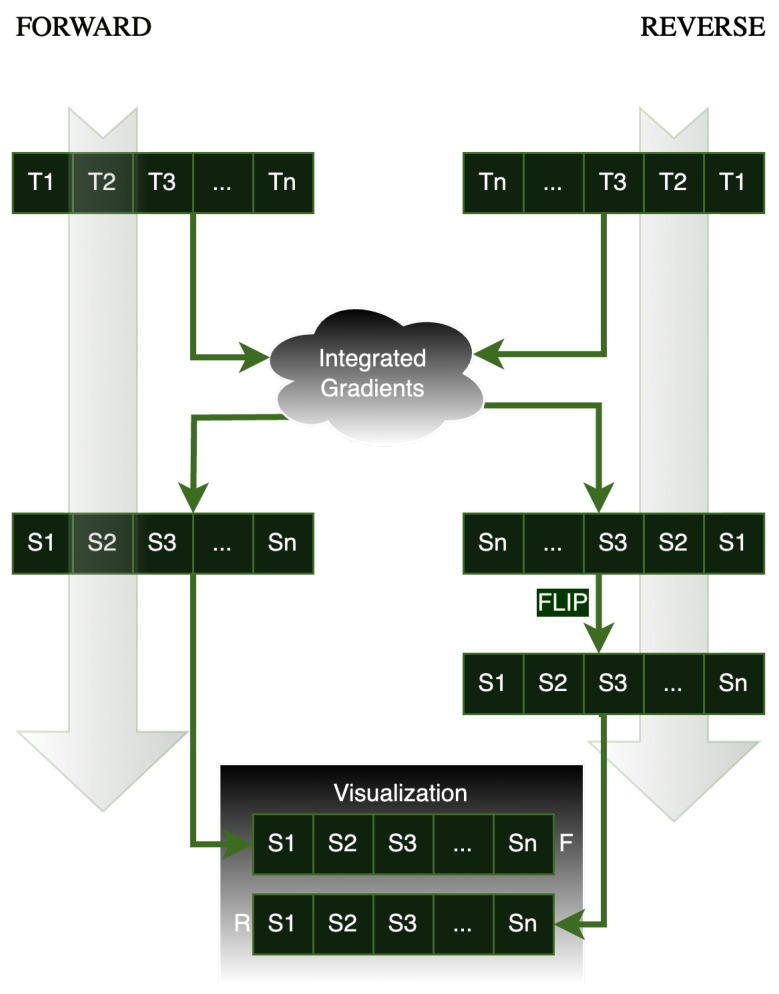
Saliency Map Visualization Method: In this method, T represents the time points, and S denotes the corresponding saliency maps for each time point. We calculate the saliency maps using integrated gradients. For the time points in the reverse pipeline, the resulting saliency maps are flipped before being presented.

**Figure 5 brainsci-15-00060-f005:**
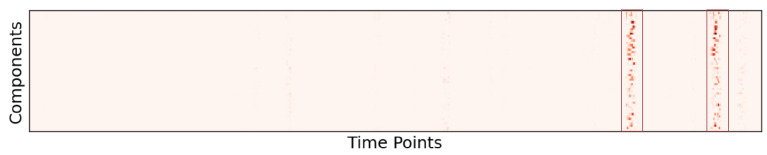
Saliency Map for a Single Subject: Temporal Dynamics and Component Importance. Saliency maps are visual representations that show which parts of the input data are most important for a model’s predictions. In the context of brain imaging data, they highlight which time points or brain regions the model considers crucial for classification. The x-axis represents time courses, while the y-axis displays components identified through ICA. Vertical red bars highlight the top 5% of features deemed most salient by the integrated gradients’ method, indicating their critical influence on model predictions.

**Figure 6 brainsci-15-00060-f006:**
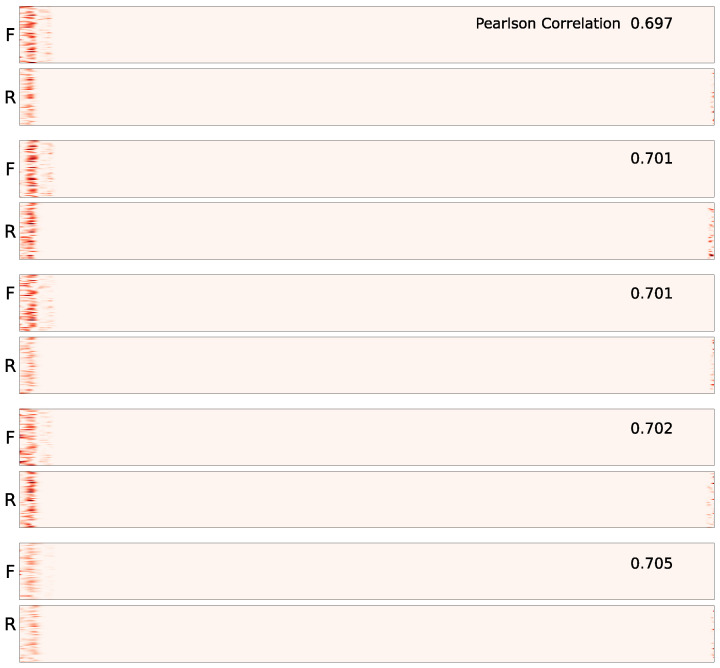
The saliency maps for five samples in the synthetic data are presented. “F” and “R” represent the saliency maps for the forward and reverse order data points, respectively. Correlation coefficient values between F and R for each sample are also presented. The red vertical bars highlight the most salient features in the data. As seen in the figure, the attributions in both the forward (F) and reverse (R) saliency maps align along the time axis for each sample.

**Figure 7 brainsci-15-00060-f007:**
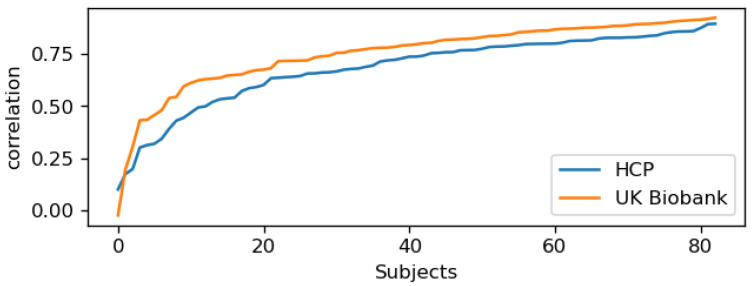
Correlation coefficient plots for the entire test sets: In both the HCP and UK Biobank datasets, the correlation is high for the majority of subjects, indicating that the vertical bars are well-aligned across forward and reverse time points for most subjects.

**Figure 8 brainsci-15-00060-f008:**
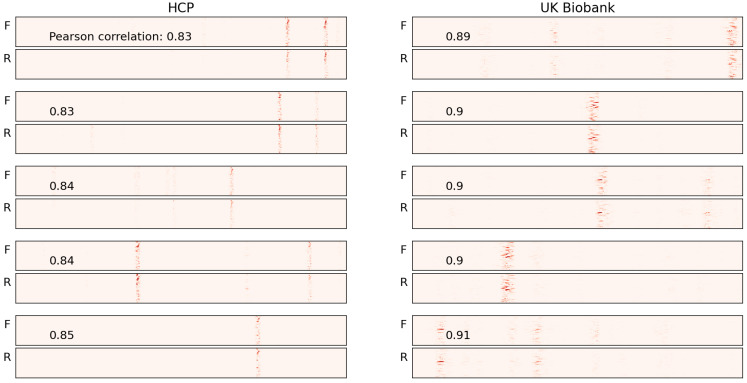
Saliency maps for five sample subjects with the highest correlation in each of the two datasets used for pretraining are presented. The correlation coefficient between the forward (F) and reverse (R) maps is shown at the top of each group. The red vertical bars highlight the most salient features that the model uses to learn the order of time points. Notably, the alignment of these bars in both the F and R maps is of particular interest. The model identifies a subset of features, and based on their position along the time axis, it distinguishes between forward and reverse time points.

**Figure 9 brainsci-15-00060-f009:**
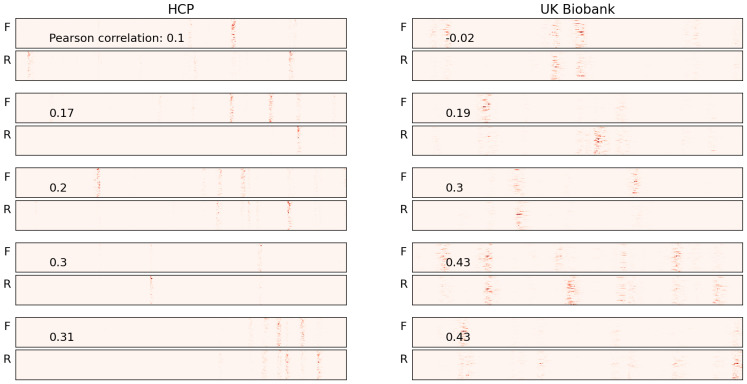
Saliency maps for five sample subjects with the lowest correlation in each of the two datasets used for pretraining are presented. In most cases, the vertical bars do not align, as reflected by the low correlation values.

**Figure 10 brainsci-15-00060-f010:**
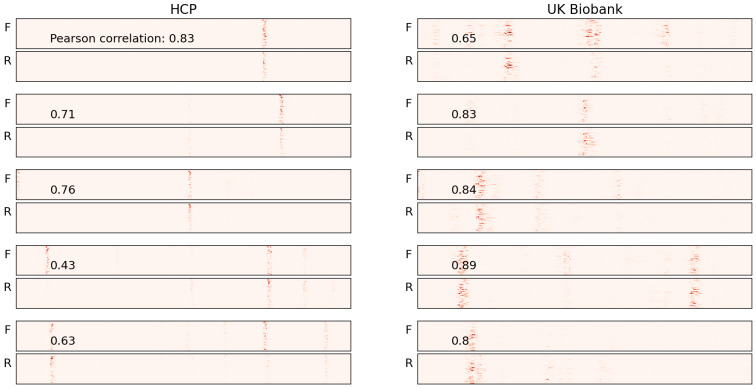
Saliency maps for five representative subjects selected using the submodular pick algorithm are shown. The submodular pick algorithm identifies subjects with high correlation values, which correspond to the alignment of salient regions in both the forward and reverse data.

**Figure 11 brainsci-15-00060-f011:**
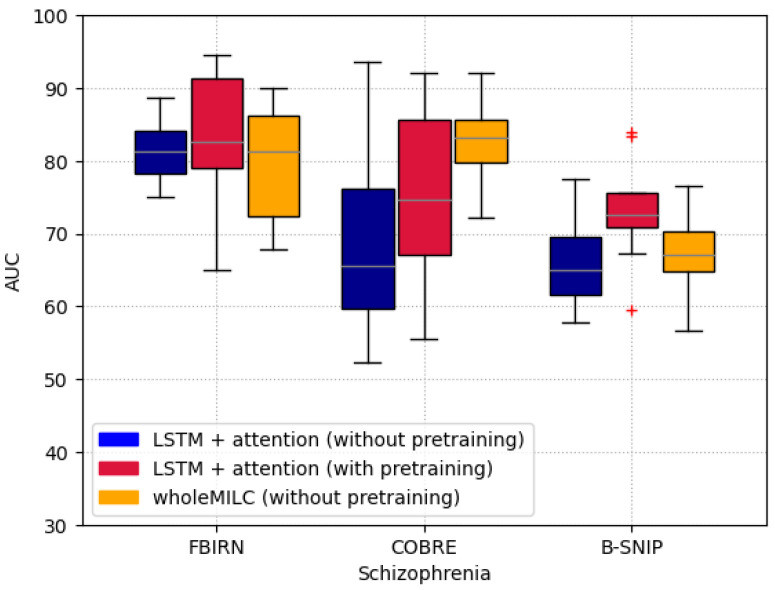
The figure illustrates the median area under the curve (AUC) scores for three different approaches in classifying schizophrenia across the FBIRN, COBRE, and B-SNIP datasets: LSTM + attention without pretraining, LSTM + attention with time reversal (TR) pretraining, and the wholeMILC model without pretraining. The red ’+’ sign in the figure represents outliers. Statistically, the median AUC for the pretrained LSTM + attention model is highest across all datasets, reaching 0.8359 on the FBIRN dataset, which signifies a significant boost in performance over its non-pretrained counterpart (median AUC of 0.7958). This suggests that pretraining with TR captures essential temporal features, enhancing the model’s ability to distinguish between schizophrenia patients and controls. The wholeMILC model, while showing a median AUC of 0.8086 on FBIRN without pretraining, does not consistently outperform the pretrained LSTM + attention model. This indicates that, despite its complexity, wholeMILC’s performance benefits less from the lack of pretraining compared to the pretrained LSTM model, with median AUC values being 0.7778 for pretrained LSTM versus 0.8254 for wholeMILC on COBRE, and 0.7224 for pretrained LSTM versus 0.6706 for wholeMILC on B-SNIP. These results underscore that pretraining significantly improves the classification capabilities of simpler architectures, making them competitive with or even superior to more complex models in terms of performance, while also enhancing interpretability.

**Figure 12 brainsci-15-00060-f012:**
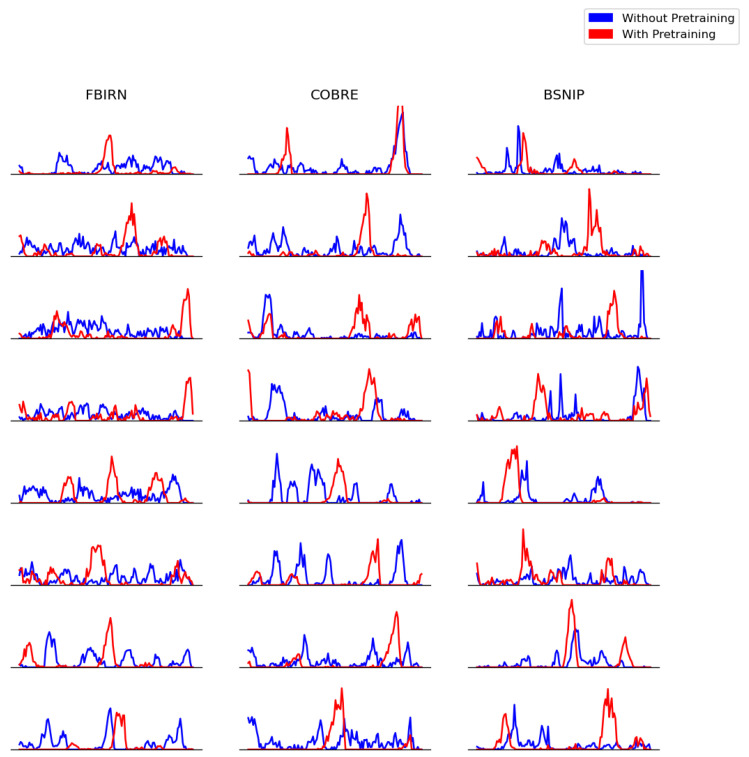
We plot the top 5% of the most salient features calculated using integrated gradients. Each row represents a subject (Schizophrenia patient), and we present the results for eight subjects from each dataset. In the majority of cases, we observe that pretraining leads to fewer spikes compared to models without pretraining, suggesting that the model identifies discriminative activity in shorter, more focused time intervals.

**Figure 13 brainsci-15-00060-f013:**
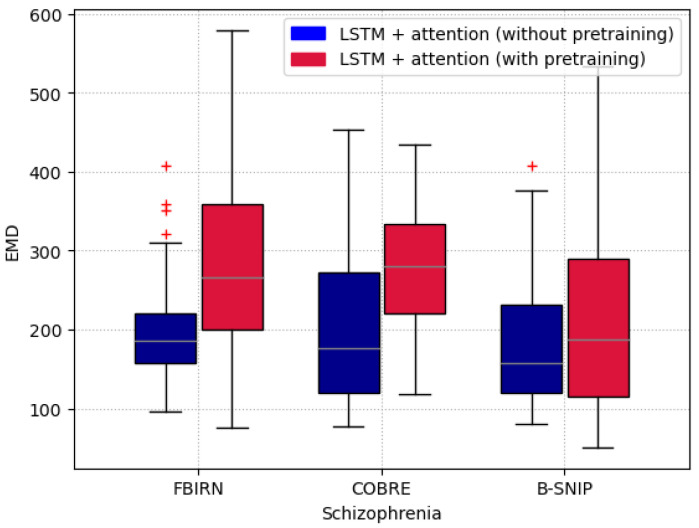
The EMD box-and-whisker plots for the downstream tasks show that higher EMD values for the models with pretraining indicate that the saliency points are concentrated into fewer spikes. This suggests that the model has identified discriminative activity within shorter, more focused time intervals. Here, The red ’+’ sign represents outliers.

**Figure 14 brainsci-15-00060-f014:**
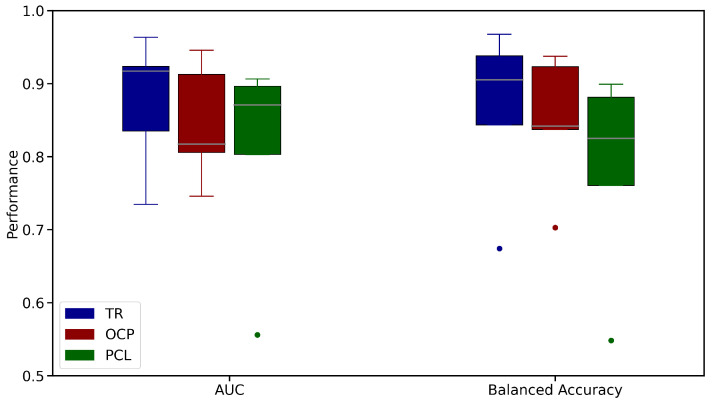
Performance Comparison of Pretraining Techniques (Time Reversal, OCP, and PCL) on Schizophrenia Classification Task using FBIRN Dataset. The dots in the figure represent outliers in box and whisker plots. Time Reversal shows the highest median AUC (0.9172) and balanced accuracy (0.9238), with the lowest variability, outperforming OCP (AUC: 0.8057, BA: 0.9128) and PCL (AUC: 0.8708, BA: 0.8965).

## Data Availability

The data utilized in this study were sourced from various public repositories and in-house datasets. These included two open-source datasets: the HCP 1200 release https://www.humanconnectome.org/study/hcp-young-adult/data-releases, accessed on 7 December 2020, and the UK Biobank https://ams.ukbiobank.ac.uk/ under Application ID 34175, initiated on 10 August 2018. Additionally, three in-house datasets were used: FBIRN https://pmc.ncbi.nlm.nih.gov/articles/PMC4651841/, COBRE https://fcon_1000.projects.nitrc.org/indi/retro/cobre.html, and B-SNIP1 https://nda.nih.gov/edit_collection.html?id=2274. All datasets underwent the NeuroMark preprocessing framework at the Center for Translational Research in Neuroimaging and Data Science (TReNDS). Due to patient data privacy regulations, the above-mentioned datasets cannot be shared publicly from TReNDS. Interested readers are encouraged to contact the respective repositories through a formal application process to request access. The NeuroMark codes have been integrated into the GIFT https://trendscenter.org/software/gift/, which is freely available for download and use by researchers worldwide. The code used for experiments presented in this study is available in our GitHub repository at Explainable_ML _TimeReversal https://github.com/zafariqballevi2/Explainable_ML_TimeReversal.git.

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
