# Peer review of "Explainable Self-Supervised Dynamic Neuroimaging Using Time Reversal"

_brainsci, 2025, doi:10.3390/brainsci15010060_

Round 1

Reviewer 1 Report

Comments and Suggestions for Authors

I have reviewed the paper titled "Explainable Self-Supervised Dynamic Neuroimaging Using Time Reversal." This work explores a novel pretraining method, Time Reversal, applied to neuroimaging data for schizophrenia classification. The focus on explainability in AI models through post-hoc analyses and the introduction of saliency maps is highly relevant. However, major revisions are required to enhance clarity, rigor, and comprehensiveness.

The following suggestions are offered for improvement:

  1. Include an ablation study to assess the impact of Time Reversal on model performance compared to alternatives. This will strengthen the claim of its efficacy.

  2. Clearly define the limitations of the proposed method, particularly in contexts of scalability and potential biases in the datasets used. Provide explicit directions for future research.

  3. Ensure consistent use and definition of abbreviations such as IG, EMD, and ICA throughout the manuscript to maintain clarity.

  4. Add a block diagram or pseudo-code to succinctly explain the methodology, especially for the pretraining and downstream classification steps.

  5. Clearly articulate the research gap addressed by this work and its contribution relative to existing literature. For example, clarify how Time Reversal surpasses other self-supervised learning methods.

  6. Include a link or repository for dataset access or code to enhance reproducibility, especially given the emphasis on pretraining with limited data.

  7. Expand the Method section to provide more detail on parameter optimization strategies and preprocessing steps, ensuring readers can replicate the experiments.

  8. Compare experimental results more extensively with prior studies, particularly with architectures such as wholeMILC, to highlight the advantages and trade-offs of your approach.

  9. Clarify the novelty of the method beyond its application to schizophrenia, addressing how it can generalize to other neuroimaging or time-series datasets.

  10. Provide a detailed explanation of how Time Reversal uniquely identifies temporal dynamics compared to existing approaches, supported by quantitative metrics.

  11. Improve commentary on tables and figures to ensure the data presented is thoroughly interpreted, particularly correlation coefficients and EMD distributions.

Comments on the Quality of English Language

The English could be improved to more clearly express the research.

Reviewer 2 Report

Comments and Suggestions for Authors

Overall Assessment

The manuscript presents a novel pretraining approach, "Time Reversal," for self-supervised learning on time-series neuroimaging data. The authors emphasize the method's application to schizophrenia classification and explainability through saliency maps and Earth Mover's Distance (EMD). The study is methodologically innovative and aligns with the increasing demand for interpretable machine learning in healthcare. However, several points require clarification, expansion, or revision to enhance the rigor and coherence of the paper.

Major Comments

1. Novelty and Contribution

  • The proposed "Time Reversal" pretraining method is described as an extension of previous work. However, the distinction between this manuscript and prior research (e.g., [19]) is insufficiently articulated. The manuscript should more clearly highlight the specific contributions and improvements over existing methods.

  • The novelty in using EMD to assess saliency maps is intriguing, but its rationale and utility compared to other metrics need a stronger theoretical foundation.

2. Experimental Design

  • Dataset Selection: While the use of diverse datasets (HCP, UK Biobank, FBIRN, COBRE, and B-SNIP) is commendable, the rationale behind selecting these datasets for both pretraining and downstream tasks is unclear. For example, why were datasets like COBRE (with a limited number of subjects) chosen for downstream tasks, and how does this choice affect model generalizability?

  • Synthetic Data: The use of synthetic chirp signals to model time-series data is creative but lacks biological plausibility. The manuscript should justify how this synthetic data contributes to understanding real-world neuroimaging signals.

3. Explainability and Interpretability

  • The explanation of saliency maps and EMD as tools for interpretability is insufficiently detailed. For instance:

    • Why were Integrated Gradients (IG) and Gradient SHAP chosen over other attribution methods?

    • How does the alignment of forward and reverse saliency maps improve interpretability?

  • The relationship between the "spikiness" of saliency maps (as quantified by EMD) and the clinical relevance of schizophrenia features requires further elaboration.

4. Comparison with Baselines

  • The comparison with wholeMILC is useful but incomplete. For instance, the manuscript should discuss why wholeMILC performs better without pretraining and what this indicates about the trade-off between model complexity and interpretability.

  • Are there other state-of-the-art methods or architectures that could serve as additional baselines? This would contextualize the performance of the proposed approach more comprehensively.

5. Statistical Validation

  • The manuscript relies heavily on AUC for evaluating performance. While this is appropriate, additional metrics (e.g., sensitivity, specificity, F1-score) would provide a more nuanced evaluation of model performance, particularly for imbalanced datasets.

  • The statistical significance of performance differences (e.g., with and without pretraining) should be assessed and reported.

6. Reproducibility

  • The code is publicly available, but key implementation details (e.g., hyperparameters, training epochs, hardware used) are missing. Including these details would enhance reproducibility.

  • Preprocessing steps, such as Independent Component Analysis (ICA), are briefly mentioned but not sufficiently detailed. Readers unfamiliar with these techniques may struggle to replicate the study.

7. Clinical Relevance

  • While the manuscript briefly mentions the clinical implications of improving schizophrenia classification, it does not explore how this method could be integrated into clinical workflows. For instance:

    • How could the interpretability offered by saliency maps support clinical decision-making?

    • Are there specific use cases where the proposed method would be most impactful?

8. Figures and Visualizations

  • Some figures (e.g., Figures 5–9) are difficult to interpret due to insufficient annotations. For instance:

    • What do the red vertical bars in the saliency maps represent in a clinical or biological context?

    • Figure 12 (EMD boxplots) lacks statistical markers (e.g., confidence intervals) to support the claims made in the text.

  • Figure 1 appears to have been sourced externally. The authors must ensure formal authorization for the use of figures from other sources and include appropriate acknowledgments.

9. Organization of Content

  • The explanation about Machine Learning methods currently in the Introduction would benefit from being moved to a separate section. This change would improve the structure and readability of the manuscript by isolating foundational knowledge from the primary narrative.

Minor Comments

Abstract

  • The abstract should explicitly mention the datasets used and summarize the key findings quantitatively (e.g., AUC scores).

Introduction

  • Clarify the distinction between interpretability and explainability, as the terms are used interchangeably throughout the manuscript.

  • Citations [1, 2, 3] on resting-state functional connectivity and deep learning are appropriate but should be supplemented with recent literature to reflect advances since 2022.

Methodology

  • The description of the LSTM and attention mechanism (Section 2.3) is too brief. A more detailed explanation, possibly with equations, would enhance comprehension.

Discussion

  • The discussion focuses heavily on the technical aspects of the model but does not sufficiently address the broader implications of the findings. For example, how does this work contribute to the growing field of explainable AI in neuroimaging?

References

  • Several references are outdated or incomplete. For instance, citation [19] appears to be the authors' prior work but is not fully described. Ensure all references are complete and formatted correctly.

Recommendations for Improvement

  1. Clarify Novelty: Distinguish the contributions of this manuscript from prior work, particularly [19].

  2. Strengthen Experimental Design: Justify dataset selection and provide more robust statistical validation.

  3. Enhance Explainability: Expand the discussion of interpretability tools and their relevance to clinical applications.

  4. Improve Visualizations: Annotate figures more thoroughly and include statistical markers where appropriate.

  5. Broaden the Discussion: Connect findings to broader themes in explainable AI and neuroimaging.

  6. Organize Content: Move the explanation of Machine Learning methods from the Introduction to a dedicated section to improve manuscript structure.

Conclusion

This manuscript addresses a significant challenge in neuroimaging—interpretable deep learning for schizophrenia classification. While the methodology is innovative, the presentation requires substantial refinement to ensure clarity, reproducibility, and alignment with the broader scientific and clinical context. Addressing the points outlined above will greatly enhance the manuscript's impact and readability.

Reviewer 3 Report

Comments and Suggestions for Authors

The manuscript presents interesting work on interpretable neuroimaging analysis but requires major revisions to strengthen the methodological description, quantitative validation, and discussion of results before publication.

1. The Introduction should consider adding a brief description of key technical concepts (e.g., saliency maps, EMD) for a wider audience.

2. Methods: (1) The time reversal pre-training method needs to explain the underlying theory/principle in more detail; (2) The architectural details such as hyperparameters, training procedures, etc. should be specified; (3) The validation methods and statistical analysis methods should be described.

3. Results presentation is generally clear but could benefit from: (1) statistical significance tests of performance differences; (2) quantitative metrics to support observations in qualitative significance plots; (3) error bars/confidence intervals on performance plots

4. Minor issues:

- Some references lack publication year

- Some plots lack clear axis labels and legends

- Minor grammatical errors throughout that need correction

- Consider adding a limitations section

Reviewer 4 Report

Comments and Suggestions for Authors

1 - Second to last sentence of the first page reads, “In contrast, Deep learning frameworks deep learning frameworks are capable…” (lines 31-32). The repetition of “deep learning frameworks” must be removed.

2 - The concept of Time Reversal (section 2.1) is critical to understanding the article. As such, its explanation must be sufficiently clear. Currently, it is only briefly outlined, and the reader is redirected to an external reference. Critical aspects of the method must be completely addressed in the present article, especially since the reference is a conference paper. Every article must be self-sustaining. This severely hinders the reader’s ability to understand the study's results and replication.

3 – In the same vein, the attention mechanism considered in section 2.3 is not sufficiently explained. The authors limit to write that they introduce an attention mechanism that “selectively focuses on the most relevant time steps by assigning dynamic weights to the input sequence, thus enabling the model to prioritize important time points while mitigating the loss of information in longer sequences”. If we, or other readers, pretend to replicate the procedure, we would not be able to do so with such a scarce description.

4 – In the “Explanations for downstream task”, section (2.6.), the concept of “distribution” within the context of the study must be explained. It is not clear what this distribution represents and which data is used, and, as such, the EMD plots cannot be appropriately interpreted by the reader.

5 - The titles of the axes must be added to the EMD plots. The spikes' meaning is unclear.

6 – In the saliency plots, the titles of the axes must also be added. Currently, it is difficult to interpret what the authors mean when they say, “the attributions in both the forward (F) and reverse (R) saliency maps align along the time axis for each sample” but then say, “the salient regions are concentrated in the lower frequency areas” (caption of figure 5). While the first sentence indicates that the horizontal axis represents time, the second is interpreted as the horizontal axis representing frequency. Therefore, the reader might interpret this as a contradiction in the analysis of the plots.

Round 2

Reviewer 1 Report

Comments and Suggestions for Authors

The authors have completely addressed all my comments, and I have no further concerns. Therefore, I recommend accepting the paper.

Comments on the Quality of English Language

The English could be improved to more clearly express the research.

Author Response

Thank you for your valuable feedback and for recommending my paper for acceptance.

Reviewer 3 Report

Comments and Suggestions for Authors

Thank you for your response to the previous reviews. While the responses are organized, a more detailed and in-depth engagement with several key points would strengthen the manuscript further. There are still some important aspects that would benefit from a more thorough discussion. I suggest minor revisions to address the following specific concerns.

1. Although a mathematical formulation for time-reversal pre-training has been added, a deeper theoretical analysis is needed to show why this method is effective in capturing temporal dependencies. The authors should add a detailed derivation process to the manuscript to enhance the rigor of the article.

2 The rationale behind the choice of hyperparameters (learning rate, batch size, etc.) should be explained. What guided these specific choices? Previous literature (please list), or how were they calculated?

3 Statistical significance tests have been added, but please state the exact statistical method, and multiple comparison corrections should be used, if not used, please explain why.

4 The differences in EMD scores are interesting, but require deeper analysis. How do these patterns relate to known temporal features of brain activity?

5. Please provide convergence analysis during training, including ablation studies of key model components
